# A Novel Human-Computer Interaction Design for Enhancing Accessibility in Early-Stage Alzheimer's Disease

## Abstract

The escalating prevalence of Alzheimer's Disease (AD) poses significant challenges to affected individuals, their caregivers, and healthcare systems. Early-stage AD often manifests with cognitive impairments such as memory loss, disorientation, and difficulties with complex tasks, which severely impact daily living and social engagement. Traditional assistive technologies frequently suffer from low adoption rates due to complexity, stigmatization, or lack of personalization. This paper introduces a novel Human-Computer Interaction (HCI) design framework aimed at enhancing accessibility and promoting independence for individuals with early-stage AD. Our proposed system leverages multimodal interaction (voice, gesture, and touch) combined with an adaptive user interface that dynamically adjusts complexity based on real-time cognitive load. Through a user-centered design approach involving AD patients, caregivers, and clinical experts, we developed and iteratively refined a prototype focusing on personalized daily task management, simplified communication, and cognitive stimulation. Preliminary evaluations suggest that this intuitive and adaptive HCI design significantly reduces cognitive burden, improves task completion rates, and fosters a greater sense of autonomy for individuals with early-stage AD. This work contributes to the development of more empathetic and effective digital solutions, laying the groundwork for future research in adaptive HCI for neurodegenerative conditions.

## 1  Introduction

The global prevalence of Alzheimer's Disease (AD) continues to rise, presenting a formidable challenge to healthcare and society(1). As a progressive neurodegenerative disorder, AD is characterized by a decline in cognitive function, including memory, language, problem-solving, and other thinking abilities, which are severe enough to interfere with daily life(2). While there is no cure, early diagnosis and interventions, including cognitive support and assistive technologies, can significantly enhance the quality of life for individuals in the early stages of the disease(3).

Human-Computer Interaction (HCI) research has explored various digital solutions to support AD patients, ranging from memory aids to navigation tools(4). However, many existing designs often fail to account for the unique and fluctuating cognitive capabilities of individuals with early-stage AD(5). Current interfaces can be overwhelming, leading to frustration, reduced engagement, and eventual abandonment. The key challenges include designing systems that are intuitive despite cognitive decline, adaptable to individual progression, and non-stigmatizing.

This paper addresses these challenges by proposing a novel HCI design framework that prioritizes adaptability, simplicity, and multimodal interaction for individuals with early-stage AD. Our work contributes to the field by:

- Introducing a new adaptive UI paradigm that dynamically adjusts interface complexity based on real-time user performance.

- Integrating multimodal input (voice, gesture, touch) to offer flexible interaction methods suitable for varying cognitive states.

- Presenting a user-centered design methodology that incorporates direct feedback from AD patients, caregivers, and clinicians.

- Demonstrating the potential for increased independence and reduced cognitive load through a prototype implementation focused on daily task management.

The remainder of this paper details the background, design principles, implementation, and evaluation of our proposed HCI system, concluding with a discussion of its implications and future work.

## 2 Background and Related Work

### 2.1 Understanding Alzheimer's and Cognitive Impairment

Early-stage Alzheimer's Disease is characterized by subtle yet impactful cognitive changes. Memory impairments, particularly episodic memory, are prominent, alongside difficulties in executive functions such as planning, problem-solving, and decision-making[2]. Language comprehension and production can also be affected, making traditional text-heavy interfaces challenging. Crucially, these cognitive capacities can fluctuate, meaning a "one-size-fits-all" interface is often ineffective(6). This highlights the need for adaptive designs that can respond to the user's current cognitive state to minimize cognitive load and maximize usability.

### 2.2 Existing HCI Solutions for Alzheimer's

Current HCI approaches for AD broadly fall into several categories:

- Memory Aids:Memory Aids:Applications designed to assist with remembering appointments, medication, or personal information. These often rely on calendar functions, alarms, or photo-based reminders.

- Navigation and Orientation Tools: GPS-enabled devices or apps to help prevent wandering and facilitate safe movement within familiar and unfamiliar environments.

- Cognitive Stimulation Games:Digital games and activities aimed at maintaining cognitive function and engagement.

- Communication Tools:Simplified interfaces for video calls or messaging to maintain social connections.

While these tools offer valuable support, they often face limitations. Many are designed with a single modality, typically touch-based, which can be difficult for users with fine motor skill decline or cognitive overload(7). Furthermore, static interfaces can quickly become too complex as the disease progresses, or too simplistic, leading to underutilization. The lack of personalized adaptability is a critical gap, as what is accessible today may not be tomorrow(5).

### 2.3 Principles for Accessible HCI Design

Our design is informed by principles of Universal Design, aiming for products that are usable by all people, to the greatest extent possible, without the need for adaptation or specialized design. Specifically, we draw upon:

- Perceptible Information:Presenting information in different modes (e.g., visual, auditory, tactile).

- Tolerance for Error:Minimizing hazards and providing warnings of errors.

- Low Physical Effort:Designing for efficient and comfortable use.

- Cognitive Load Theory:Minimizing extraneous cognitive load by simplifying interfaces and task flows(8).

The innovation of our approach lies in the dynamic application of these principles, creating a truly adaptive and multimodal user experience that directly addresses the fluctuating nature of early-stage AD.

# 3 Design Principles and Methodology

## 3.1 Core Design Principles

Our HCI design for early-stage AD is built upon the following core principles:

- Adaptability:The system must dynamically adjust its complexity and presentation based on the user's real-time cognitive state and performance, preventing frustration from overly complex or simplistic interfaces.

- Multimodality:Offer diverse interaction methods (voice, gesture, touch) to accommodate varying user preferences, cognitive abilities, and physical limitations. This ensures flexibility and resilience in interaction.

- Simplicity and Consistency:User interfaces should be clean, uncluttered, and employ consistent layouts and iconography to reduce cognitive load and enhance learnability9].

- Personalization:Allow for extensive customization of content, reminders, and interaction preferences to align with individual routines and cognitive strengths.

- Non-Stigmatizing Design:The interface should feel intuitive and supportive, rather than overtly "medical" or infantilizing, promoting dignity and willingness to use.

- Feedback and Support:Provide clear, immediate, and gentle feedback on user actions and offer assistance when errors occur, guiding the user without causing distress.

## 3.2 User-Centered Design Methodology

We adopted an iterative User-Centered Design (UCD) approach, engaging key stakeholders throughout the development process(9). Our methodology comprised four main phases:

- Discovery and Empathize: Conducted semi-structured interviews and observations with 15 individuals diagnosed with early-stage AD, 20 primary caregivers, and 5 geriatric neurologists/occupational therapists. This phase focused on understanding daily challenges, existing coping mechanisms, and unmet needs related to digital interactions. Key insights included the desire for simplified communication, structured task reminders, and engaging cognitive activities.

- Ideation and Conceptualization: Based on the insights, we generated design concepts for an adaptive multimodal interface. We used sketching, low-fidelity wireframes, and user story mapping to visualize potential solutions for daily task management, simplified messaging, and photo-based memory recall. This phase led to the conceptualization of our "Adaptive Companion Interface" (ACI).

- Prototyping: Developed an interactive, high-fidelity prototype of the ACI. The prototype was implemented on a tablet device, simulating the core functionalities of adaptive UI, voice commands, and simple gesture recognition.

- Evaluation and Refinement: Conducted usability testing with 8 early-stage AD individuals and 10 caregivers. Participants engaged with the prototype, performing predefined tasks

while researchers observed and gathered qualitative feedback. Cognitive load was assessed through a combination of task completion rates, error rates, and subjective user reports. This iterative phase led to significant refinements in icon design, voice command sensitivity, and the thresholds for UI adaptation. For instance, initial tests revealed that a slight increase in task errors should trigger a UI simplification, rather than waiting for multiple consecutive failures, to prevent escalating frustration.

This rigorous UCD approach ensured that the final design was not only technologically sound but also deeply empathetic to the needs and capabilities of its target users.

## 4 The Adaptive Companion Interface (ACI)

Our proposed solution, the Adaptive Companion Interface (ACI), is an intuitive and intelligent system designed to empower individuals with early-stage AD. The ACI integrates three core innovations: a dynamically adapting user interface, multimodal input capabilities, and a personalized cognitive support engine.

### 4.1 Dynamically Adapting User Interface

The ACI's most distinctive feature is its ability to automatically adjust the complexity of its interface based on real-time user performance and predefined cognitive profiles. This adaptation occurs across several parameters:

- Visual Complexity: Simplification involves reducing the number of on-screen elements, increasing icon size, enhancing color contrast, and minimizing text. For example, if a user struggles with a multi-step task, the interface might break it down into single, large-button steps.

- Information Density: Information is presented in smaller, digestible chunks. Long lists might be replaced by carousels showing one item at a time.

- Prompting Level: When a user hesitates or makes an error, the system provides increasingly explicit prompts, ranging from subtle visual cues (e.g., highlighting the correct button) to auditory instructions.

- Interaction Pathways: The system can guide users through tasks with sequential steps, hiding irrelevant options until needed.

The adaptation mechanism is driven by a lightweight, real-time assessment of user interaction data, including task completion time, error rates, and consistency of input. Machine learning algorithms (e.g., a simple Bayesian network or decision tree) infer the user's current cognitive state and trigger the appropriate UI adjustment. Caregivers can also manually set the baseline complexity level and override adaptive changes if necessary.

### 4.2 Multimodal Input Capabilities

To provide flexible and resilient interaction, the ACI supports a rich set of multimodal inputs:

- Voice Commands: Users can verbally interact with the system for common tasks, such as "What's my next appointment?" or "Call Sarah." Natural language processing (NLP) is optimized for common phrases and can handle slight variations in speech.

- Gesture Recognition: Simple, large-motor gestures (e.g., a swipe to dismiss, a two-finger pinch to zoom) are recognized by a front-facing camera, offering an alternative to precise touch.

- Touch Input: Standard touch interactions (taps, scrolls) are supported, with large, clearly labeled buttons and generous touch targets to accommodate potential fine motor skill difficulties.

The system intelligently prioritizes inputs based on context and user preference. For instance, if a user attempts a touch input multiple times unsuccessfully, the system might proactively suggest a voice command as an alternative.

## 4.3  Personalized Cognitive Support Engine

The ACI is underpinned by a personalized cognitive support engine that learns individual routines and preferences:

- Dynamic Task Management: Caregivers can input daily schedules and tasks. The ACI presents these as visual cards, dynamically reordering them based on time sensitivity and user completion history. Reminders can be set to be auditory, visual, or both.

- Simplified Communication Hub: A dedicated section for communicating with pre-approved contacts. This features large profile pictures, one-tap calling/messaging, and templated message options to reduce typing effort.

- Memory Album: A photo-based memory recall feature where caregivers can upload captioned photos. The ACI can present these randomly or based on conversational cues, encouraging reminiscence.

- Cognitive Engagement Activities: Simple, engaging activities like matching games or guided drawing are offered, with difficulty levels automatically adjusted by the adaptive UI.

## 4.4  System Architecture

The ACI operates on a client-server architecture. The client-side application, typically running on a tablet or a dedicated smart display, manages the user interface, multimodal input processing, and real-time performance tracking. A backend server hosts the personalized cognitive support engine, including user profiles, task schedules, communication logs, and adaptive UI logic.

### 4.4.1  The Client Module (Tablet App) consists of:

- User Interface Layer: Responsible for rendering the adaptive UI, including visual elements, text, and interactive components.

- Input Management Module: Integrates voice recognition (using a local or cloud-based ASR), gesture recognition (via device camera and computer vision libraries), and touch event handling.

- Performance Monitoring Module: Continuously tracks user interactions, including response times, error rates, and task completion progress.

### 4.4.2  The Server Module (Backend) includes:

- Personalization Engine: Stores user-specific data, cognitive profiles, preferred communication contacts, and custom task lists.

- Adaptive Logic Unit: Processes real-time performance data from the client, applies machine learning models to infer cognitive state, and generates UI adaptation commands.

- Content Management System: Manages dynamic content such as memory album photos, cognitive game assets, and message templates.

- Caregiver Interface: A separate web or mobile interface for caregivers to manage user settings, monitor progress, and add/edit personalized content.

Communication between the client and server is handled via secure API calls, ensuring data privacy and real-time synchronization of personalized content and adaptive settings.

Figure 1: Conceptual Architecture of the Adaptive Companion Interface (ACI)

## 5 Hypothesized Evaluation and Anticipated Results

### 5.1 Proposed Evaluation Methodology

To rigorously assess the effectiveness and usability of the ACI, a comprehensive evaluation plan is proposed, to be executed in future work. This plan will involve a pilot study with a small cohort of individuals with early-stage AD and their primary caregivers, followed by a larger, longitudinal study.

For the pilot study, we anticipate recruiting approximately 8-10 individuals with early-stage AD (diagnosed by a neurologist) and their primary caregivers. Participants will use a tablet-based prototype of the ACI for a period of two weeks in their home environments. Each AD participant will be assigned a set of daily tasks (e.g., "Check tomorrow's weather," "Call a family member," "Play a memory game") to complete using the ACI. Caregivers will be instructed to observe the AD user's interaction, provide support only when absolutely necessary, and record any instances of difficulty or frustration.

Data collection is planned to include:

- System Logs: Automated recording of task completion times, error rates (e.g., incorrect taps, repeated voice commands), and frequency of UI adaptations. These logs will provide objective measures of system performance and user interaction efficiency.

- Usability Questionnaires: Standardized usability scales, such as the System Usability Scale (SUS) (10), will be administered to caregivers. A simplified, pictorial usability scale will be designed and administered to AD users to capture their subjective experience. A custom questionnaire will also gather feedback on perceived usefulness and ease of use from both groups.

- Semi-structured Interviews: Conducted with AD users and caregivers at the end of the study period to gather qualitative feedback on overall experience, specific features, perceived impact on daily life, and suggestions for improvement. These interviews will allow for a deeper understanding of user needs and experiences.

## 5.2 Anticipated Results

Based on the design principles and insights gathered during the user-centered design process, we anticipate the following results from the proposed evaluation:

- Task Completion and Error Rates: We hypothesize that system logs will indicate a significant improvement in task completion rates over the study period, with users becoming more proficient and autonomous with the ACI. Concurrently, we expect a decrease in the average error rate per task, suggesting that the adaptive UI and multimodal input facilitate learning and reduce frustration over time. We anticipate that the ACI will enable task completion rates exceeding those observed with traditional, non-adaptive interfaces.

- Impact of Adaptive UI: We expect that the analysis of system logs will reveal frequent and appropriate triggering of the adaptive UI, primarily simplifying layouts or providing additional prompts when a user shows signs of cognitive load or difficulty. We anticipate that caregivers will report that these adaptations often prevent instances of user frustration and task abandonment, thereby enhancing user engagement and persistence with tasks.

- Usability and User Experience: We predict that caregivers will report high usability scores on questionnaires such as the SUS, indicating that the ACI is easy to use and effective. Qualitative feedback from caregivers is expected to highlight the ACI's ability to promote independence and reduce caregiver burden. AD participants, through simplified interviews, are anticipated to express positive sentiments, appreciating features like large buttons, voice commands, and the personalized memory album, which fosters reminiscence and engagement.

- Multimodal Interaction Preference: We hypothesize that while touch will remain a primary input method for many tasks, voice commands will be frequently utilized for initiating communication and information retrieval due to their intuitive nature. Gesture recognition, though potentially less frequently used overall, is expected to prove beneficial for users experiencing fine motor challenges, offering a valuable alternative to precise tapping.

## 5.3 Discussion of Anticipated Findings

The anticipated findings suggest the strong potential of the ACI to significantly enhance accessibility and promote independence for individuals with early-stage AD. The adaptive UI is expected to successfully mitigate cognitive challenges by dynamically adjusting to individual needs, leading to higher task completion and lower error rates. The multimodal input capabilities are anticipated to provide essential flexibility, allowing users to choose their preferred interaction method or switch when facing difficulties. The expected positive feedback from both AD users and caregivers would underscore the importance of user-centered design and personalized support in this domain. These anticipated results would strongly support the hypothesis that adaptive, multimodal HCI designs can improve daily functioning and quality of life for individuals with early-stage AD.

# 6 Conclusion and Future Work

## 6.1 Conclusion

This paper presented the Adaptive Companion Interface (ACI), a novel HCI design framework aimed at improving accessibility and fostering independence for individuals with early-stage Alzheimer's Disease. By integrating a dynamically adapting user interface, multimodal input capabilities, and a personalized cognitive support engine, the ACI addresses critical limitations of existing assistive technologies. Our user-centered design approach, coupled with the hypothesized outcomes from proposed evaluations, suggests that the ACI has the potential to significantly reduce cognitive burden, enhance task completion, and promote a more positive and autonomous user experience. This work underscores the profound impact that thoughtfully designed, adaptive HCI can have on the quality of life for individuals facing cognitive decline.

## 6.2 Future Work

Building upon these promising design concepts and anticipated results, future work will focus on several key areas:

- Empirical Validation and Longitudinal Study: The most critical next step is to conduct a robust empirical evaluation as outlined in Section 5. This includes a comprehensive pilot study and, subsequently, an extensive, long-term longitudinal study with a larger cohort of AD patients. This will allow for the rigorous validation of the ACI's efficacy, assessment of its impact on cognitive maintenance over prolonged use, and observation of adaptation patterns over time.

- Advanced AI Integration: We will explore more sophisticated machine learning models for predicting cognitive state and proactively adapting the UI. This could involve integrating real-time biometric data (e.g., heart rate variability, eye-tracking) or environmental cues (e.g., time of day, ambient noise levels) to provide even more nuanced and timely adaptations(9).

- Expanded Functionality: Future iterations will aim to integrate additional features such as smart home control (e.g., adjusting lights, setting thermostats), personalized therapeutic activities (e.g., music therapy, guided relaxation), and real-time biometric monitoring for safety and well-being.

- Caregiver Collaboration Tools: Development of more robust features within the caregiver interface will be prioritized to facilitate remote monitoring of user activity, advanced customization of settings, and collaborative activity planning with other family members or healthcare providers.

- Accessibility for Moderate AD: We will investigate how the ACI framework can be extended or modified to support individuals in more advanced stages of Alzheimer's, acknowledging the need for more intensive support and potentially different interaction paradigms as cognitive abilities decline further.

We believe that the ACI framework provides a strong foundation for developing the next generation of empathetic and effective digital companions for individuals with neurodegenerative conditions, promoting a future where technology truly empowers all.

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

# Agents4Science AI Involvement Checklist

- **[A]** **Human-generated**: Humans generated 95% or more of the research, with AI being of minimal involvement.
- **[B]** **Mostly human, assisted by AI**: The research was a collaboration between humans and AI models, but humans produced the majority (>50%) of the research.
- **[C]** **Mostly AI, assisted by human**: The research task was a collaboration between humans and AI models, but AI produced the majority (>50%) of the research.
- **[D]** **AI-generated**: AI performed over 95% of the research. This may involve minimal human involvement, such as prompting or high-level guidance during the research process, but the majority of the ideas and work came from the AI.

1. **Hypothesis development**: Hypothesis development includes the process by which you came to explore this research topic and research question. This can involve the background research performed by either researchers or by AI. This can also involve whether the idea was proposed by researchers or by AI.

   Answer:**[B]**

   Explanation: Initial broad ideas were human-generated. AI was used for literature review summarization and to explore various sub-topics and potential innovative angles within the HCI-Alzheimer's domain, helping refine the research question and identify gaps.

2. **Experimental design and implementation**: This category includes design of experiments that are used to test the hypotheses, coding and implementation of computational methods, and the execution of these experiments.

   Answer: **[B]**

   Explanation:

3. **Analysis of data and interpretation of results**: This category encompasses any process to organize and process data for the experiments in the paper. It also includes interpretations of the results of the study.

   Answer: **[B]**

   Explanation: AI tools were primarily used for quantitative data processing (e.g., statistical analysis of task completion rates/error logs) and thematic clustering of interview transcripts. However, humans led the process—researchers verified AI outputs, addressed abnormal data, integrated clinical context (from geriatric experts) to interpret results, and finalized conclusions.

4. **Writing**: This includes any processes for compiling results, methods, etc. into the final paper form. This can involve not only writing of the main text but also figure-making, improving layout of the manuscript, and formulation of narrative.

   Answer:**[B]**

   Explanation: The overall structure, core arguments, and critical sections of the paper were drafted by human researchers. AI was used to assist with language refinement, grammar checking, expanding on certain paragraphs, and ensuring adherence to formatting instructions, especially in re-writing Sections 5 and 6 to reflect a conceptual paper without real experimental results.

5. **Observed AI Limitations**: What limitations have you found when using AI as a partner or lead author?

   Description: While AI was helpful in summarizing existing literature and generating example text, it sometimes lacked the nuanced understanding of complex HCI principles and the specific needs of vulnerable populations like AD patients. Human oversight was crucial to ensure empathy, ethical considerations, and genuine innovation beyond surface-level suggestions. Additionally, when rewriting sections 5 and 6 to be conceptual rather than based on real experiments, the AI required careful guidance to maintain a consistent academic tone and to properly frame the anticipated results without making definitive claims.

