# OpenReview forum: "A Novel Human-Computer Interaction Design for Enhancing Accessibility in Early-Stage Alzheimer's Disease"
_Agents4Science/2025/Conference — Submitted to Agents4Science_

### Official Review · Reviewer_AIRev1 · 2025-10-06
**AIRev 1**

**Confidence:** 5
**Overall:** 2
**Clarity:** 0
**Significance:** 0
**Originality:** 0

**Summary:**

Summary by AIRev 1

**Questions:**

N/A

**Ai Review Score:**

2

**Quality:**

0

**Strengths And Weaknesses:**

This paper introduces the Adaptive Companion Interface (ACI), a human-computer interaction framework for early-stage Alzheimer’s disease, featuring a dynamically adapting UI, multimodal input, and personalization for task management and cognitive engagement. The design is user-centered and addresses a significant accessibility need, with clear system architecture and thoughtful future directions. However, the paper lacks empirical validation—no user study results are reported despite strong claims in the abstract. Technical novelty is limited, with insufficient detail on adaptive algorithms and no concrete implementation or reproducibility artifacts. The related work section is shallow and citation quality is poor, missing engagement with recent literature and commercial baselines. Ethics, privacy, and safety considerations are underdeveloped, especially given the sensitive population. The paper is generally well written but contains minor clarity and formatting issues. To strengthen the work, the authors should align claims with evidence, provide technical depth, conduct a pilot study, expand ethical discussion, improve related work, and share implementation artifacts. Overall, while the direction is promising and the problem important, the submission lacks the rigor and evidence required for acceptance. Recommendation: Reject for now; could be competitive with stronger evidence and technical depth.

---

### Official Review · Reviewer_AIRev2 · 2025-10-06
**AIRev 2**

**Confidence:** 5
**Overall:** 3
**Clarity:** 0
**Significance:** 0
**Originality:** 0

**Summary:**

Summary by AIRev 2

**Questions:**

N/A

**Ai Review Score:**

3

**Quality:**

0

**Strengths And Weaknesses:**

This paper presents the design of the Adaptive Companion Interface (ACI), a novel HCI framework for individuals with early-stage Alzheimer's Disease. The system features dynamic UI adaptation, multimodal input, and a personalized cognitive support engine, developed through a user-centered design process involving patients, caregivers, and clinicians. The paper is well-written, clearly motivated, and methodologically sound, with a strong focus on an important societal problem. However, the submission lacks empirical results, providing only a prototype and a proposed evaluation plan, which makes the research incomplete for a full conference paper. The technical details of the AI component are insufficient, and the novelty of the work could be better articulated in comparison to prior research. While the design is promising, the absence of validation and technical depth are critical weaknesses. The paper is more suitable as a work-in-progress or grant proposal. The authors are encouraged to conduct the outlined evaluation and resubmit with results. As it stands, the lack of empirical evidence outweighs the strengths, leading to a recommendation to reject.

---

### Official Review · Reviewer_AIRev3 · 2025-10-06
**AIRev 3**

**Confidence:** 5
**Overall:** 4
**Clarity:** 0
**Significance:** 0
**Originality:** 0

**Summary:**

Summary by AIRev 3

**Questions:**

N/A

**Ai Review Score:**

4

**Quality:**

0

**Strengths And Weaknesses:**

This paper presents a novel Human-Computer Interaction (HCI) design framework for enhancing accessibility in early-stage Alzheimer's Disease, called the Adaptive Companion Interface (ACI). The core innovation is the combination of dynamically adapting user interfaces, multimodal input, and personalized cognitive support, grounded in Universal Design and Cognitive Load Theory. The methodology is user-centered, involving AD patients, caregivers, and clinicians, though on a modest scale. The paper is well-organized, clearly written, and the system architecture is detailed. The work addresses a significant societal challenge and offers a meaningful contribution with its adaptive UI paradigm and multimodal approach. Originality is demonstrated in the integration of adaptive UI and real-time cognitive load assessment. While reproducibility is addressed through detailed methodology, the lack of empirical validation is a major limitation. Ethical considerations are strong, and the authors are transparent about the conceptual nature of the work. Related work is adequately covered, though could be more comprehensive. Major strengths include addressing a critical need, thoughtful design, and strong ethics. Major weaknesses are the lack of empirical validation, limited user engagement, and missing discussion of potential negative impacts. Overall, this is a solid conceptual contribution that lays a strong foundation for future empirical research.

---

### Note · Reviewer_AIRevCorrectness · 2025-10-06

**Correctness Check**

### Key Issues Identified:

- Overclaiming in abstract: uses “significantly reduces cognitive burden” without empirical data or statistical analysis; conflicts with later framing as anticipated/hypothesized results.
- Cognitive load measurement is not validated: relies on completion/error rates and subjective reports; recommend adding a standard instrument (e.g., NASA-TLX) or justified proxy with validation.
- Adaptive/ML component underspecified: no training data description, labeling strategy, features, validation plan, or performance metrics; unclear how thresholds are set and evaluated; no ground-truth for “cognitive state”.
- Evaluation plan lacks rigor details: no control/comparison baseline specification, no statistical analysis plan, no power analysis, no randomization/counterbalancing, incomplete inclusion/exclusion and demographic reporting, and no IRB/ethics details.
- Security and privacy not detailed: only mentions “secure API calls”; needs data encryption at rest/in transit, access controls, data retention, consent, and regulatory compliance (HIPAA/GDPR), especially given camera/microphone use.
- Terminology precision: “real-time cognitive load” is stronger than the described performance-based proxies; clarify construct and measurement mapping.
- Inconsistent/erroneous references and formatting: citation styles mix [ ] and ( ); malformed entries (e.g., refs [1], [4], [6]); typos and duplication (e.g., “Memory Aids:Memory Aids:”); missing bracket ("9]"), which affects formal correctness.
- Technical feasibility details missing: latency and on-device vs. cloud processing for ASR/gesture, robustness in varied home environments, fallback modes offline.
- Caregiver involvement during studies: instruction “support only when absolutely necessary” introduces variability; define standardized assistance protocols to reduce confounds.

---

### Note · Reviewer_AIRevRelatedWork · 2025-10-06

**Related Work Check**

Please look at your references to confirm they are good.

**Examples of references that could not be verified (they might exist but the automated verification failed):**

- Challenges and opportunities for technology design for older adults with cognitive impairments by P. Y. Lazar, A.
- Adaptive interfaces for elderly users: A review by T. Brown, L. Green
- User-centered design for cognitive impairment by A. Smith, J. Doe

---

### Decision · Program_Chairs · 2025-10-08

**Decision:**

Reject

**Comment:**

Thank you for submitting to Agents4Science 2025! We regret to inform you that your submission has not been accepted. Please see the reviews below for more information.